# A  APPENDIX

In this Appendix, we provide additional details and supplementary analyses to complement the main paper. Specifically, we first describe the experimental settings in greater detail (Section A.1), followed by supplementary discussions of the methodology (Section A.3). We then elaborate on the design and training of the manifold reconstructor (Section A.4), and finally present the implementation details and motivation of the random projection used in our ablation study (Section A.5). These materials are intended to provide deeper technical insights, clarify design choices, and ensure reproducibility of our results.

## A.1  EXPERIMENT SETTINGS IN DETAIL

**Model Architecture.**    We implement our model using a conditional latent diffusion framework. The denoising network is a 12-layer Transformer with pre-LayerNorm, model dimension 768, 12 attention heads, and feed-forward dimension 3072. Cross-attention is applied over sentence-level embeddings from a frozen BART encoder.

**Semantic Safety Manifold.**    To construct the safety manifold, we use latent representations of verified non-toxic sentences (from Detoxify and Jigsaw), reduced via diffusion maps (with kernel $\epsilon = 0.1$ and 50 eigenvectors). During generation, we project latent states back onto this manifold every 5 steps to avoid semantic drift into harmful regions.

**Training Configuration.**    Training uses 1000 denoising steps with a cosine noise schedule. We use classifier-free guidance with 10% conditioning dropout. Optimization uses AdamW (learning rate $1 \times 10^{-4}$, batch size 32) for 200K steps, with gradient clipping (1.0) and mixed-precision (fp16). Training runs on 2 A40 GPUs for 48 hours.

**Inference.**    At inference, we apply CFG with scale 2.0 and project latent states to the manifold every 5 steps. The final latent is decoded via a BART decoder using greedy decoding.

## A.2  CASES STUDY

Additional case study examples are presented in Table 4.

## A.3  METHODOLOGY SUPPLEMENT

### A.3.1  CONSTRUCTING THE DIFFUSION KERNEL

A key step in modeling the safe semantic manifold $\mathcal{M}_{\text{safe}} \subset \mathbb{R}^{d_m}$ is to capture the intrinsic geometry of non-offensive latent representations, beyond what is observable through naive Euclidean distances. To this end, we adopt the diffusion maps framework, which begins by constructing a heat diffusion kernel to define local connectivity and semantic similarity across latent points.

Let $\{\mathbf{z}^i \in \mathbb{R}^d\}_{i=1}^N$ denote the set of latent vectors corresponding to safe (i.e., non-offensive) sentences. Instead of directly using pairwise Euclidean distances, we define a diffusion kernel $K \in \mathbb{R}^{N \times N}$ via a Gaussian similarity function:

$$K_{ij} = \exp\left(-\frac{\|\mathbf{z}^i - \mathbf{z}^j\|^2}{\delta}\right), \tag{11}$$

where $\delta > 0$ is a bandwidth parameter controlling the local sensitivity. This kernel quantifies how likely two semantic representations $\mathbf{z}_i$ and $\mathbf{z}_j$ are to "communicate" under a simulated diffusion process. In effect, $K$ encodes not just distance, but connectivity along the (potentially curved) semantic manifold.

We then row-normalize the kernel to obtain a Markov transition matrix $P$, such that each row sums to one:

$$P_{ij} = \frac{K_{ij}}{\sum_k K_{ik}}. \tag{12}$$

This transition matrix defines a random walk over the latent points, where transitions are more probable between semantically similar sentences. Crucially, the structure of $P$ captures the global geometry of $\mathcal{Z}_{\text{safe}}$, and facilitates the definition of a diffusion distance between latent vectors:

$$d_t(\mathbf{z}_i, \mathbf{z}_j)^2 = \sum_{k=1}^{N} \lambda_k^{2t} \left( \psi_k(i) - \psi_k(j) \right)^2, \tag{13}$$

where $\{(\lambda_k, \psi_k)\}$ are the eigenpairs of $P$, and $t$ controls the diffusion timescale. This distance measures how easily information can propagate between two points over the graph induced by the kernel, which aligns well with human semantic perception.

Compared to raw distances in $\mathbb{R}^d$, diffusion distance is robust to noise and more sensitive to underlying nonlinear structures, especially when the safe latent distribution lies on a lower-dimensional manifold embedded in high-dimensional space.

The primary benefits of constructing the diffusion kernel are thus threefold:

Geometry-aware semantic similarity: Captures how semantically similar two points are based on their mutual accessibility through local neighborhoods, rather than raw vector distance. Spectral embedding foundation: Enables low-dimensional diffusion coordinates $\Psi(\mathbf{z})$ via eigendecomposition of $P$, preserving manifold topology. Manifold-constrained rewriting: Facilitates explicit projection of latent diffusion trajectories onto the safe region $\mathcal{M}_{\text{safe}}$, either directly or via gradient-based correction.

These properties make diffusion kernel construction a foundational step in enabling safe and semantically faithful rewriting within our framework. The subsequent manifold reconstructor $\phi(\cdot)$ learns to decode these diffusion coordinates back into valid latent vectors, completing the mapping $\phi \circ \Psi : \mathbb{R}^d \to \mathcal{M}_{\text{safe}} \subset \mathbb{R}^d$.

In Equation (2), the Markov transition matrix $P \in \mathbb{R}^{N \times N}$ is constructed by row-normalizing the diffusion kernel $K$, with its elements defined as:

$$P_{ij} = \frac{K_{ij}}{\sum_k K_{ik}}. \tag{14}$$

$i \in \{1, \ldots, N\}$ indexes the source latent vector $\mathbf{z}_i$, representing the current point in the semantic latent space; $j \in \{1, \ldots, N\}$ indexes the target latent vector $\mathbf{z}_j$, representing a potential next step in the diffusion process; $k \in \{1, \ldots, N\}$ enumerates all possible latent vectors for normalization, ensuring that each row of $P$ sums to 1, i.e., $\sum_j P_{ij} = 1$.

Intuitively, $P_{ij}$ reflects the probability of transitioning from latent vector $\mathbf{z}^i$ to $\mathbf{z}^j$ under a single-step random walk over the semantic graph induced by the heat kernel. The transition probabilities are higher between semantically similar sentences, as encoded by the Gaussian affinity $K_{ij}$. This probabilistic interpretation allows us to define a diffusion process on the data, which serves as a proxy for measuring semantic connectivity along the intrinsic manifold.

Such a construction enables us to go beyond raw Euclidean distances, capturing the true semantic geometry of safe latent representations, and supports downstream spectral embedding, projection, and constrained sampling operations.

### A.3.2 LEARNING THE DIFFUSION COORDINATES

Once the transition matrix $P \in \mathbb{R}^{N \times N}$ is constructed, we proceed to learn the diffusion coordinates—a nonlinear, low-dimensional embedding of the latent vectors that preserves the intrinsic geometry of the data manifold. This is achieved via spectral decomposition of the Markov matrix:

$$P \psi_k = \lambda_k \psi_k, \quad k = 1, 2, \ldots, N, \tag{15}$$

where $\lambda_k \in [0, 1]$ and $\psi_k \in \mathbb{R}^N$ denote the eigenvalues and corresponding right eigenvectors of $P$. These eigenvectors define an orthonormal basis over the semantic latent space, ordered by decreasing eigenvalues.

The diffusion map of a latent vector $\mathbf{z}^i$ is then defined as:

$$\Psi(\mathbf{z}^i) = \left( \lambda_1^t \psi_1(i), \; \lambda_2^t \psi_2(i), \; \ldots, \; \lambda_m^t \psi_m(i) \right) \in \mathbb{R}^m, \tag{16}$$

where $m$ is the target embedding dimensionality, and $t$ is the diffusion time scale controlling the influence of global versus local structure.

In the equation above, $\psi_k(i)$ denotes the $i$-th component of the $k$-th eigenvector, i.e., the value of the $k$-th eigenfunction at the data point $\mathbf{z}^i$; $\lambda_k^t$ is the eigenvalue raised to the power $t$, which exponentially downweights less significant directions and filters out noise; $\Psi(\mathbf{z}^i) \in \mathbb{R}^m$ is the diffusion coordinate vector of $\mathbf{z}^i$, representing its position on the intrinsic manifold.

Each coordinate $\Psi_k(\mathbf{z}^i) = \lambda_k^t \psi_k(i)$ captures how information diffuses from $\mathbf{z}_i$ along the $k$-th principal direction of the semantic graph. Lower-indexed components (with higher $\lambda_k$) reflect more stable, slowly varying semantic structures, while higher-indexed components tend to capture fine-grained or noisy variations.

This coordinate system defines a nonlinear low-dimensional chart for the safe semantic manifold $\mathcal{M}_{\text{safe}}$, where distances between points approximate the diffusion distance:

$$d_t(\mathbf{z}^i, \mathbf{z}^j)^2 \approx \left\| \Psi(\mathbf{z}^i) - \Psi(\mathbf{z}^j) \right\|^2. \tag{17}$$

Thus, the diffusion embedding $\Psi(\cdot)$ serves two critical purposes: It provides a compact and geometry-aware latent representation of safe text, and it enables the construction of a decodable safe manifold via the mapping $\phi : \mathbb{R}^m \to \mathbb{R}^d$, which we learn in the next stage.

### A.4 LEARNING THE MANIFOLD RECONSTRUCTOR

The diffusion coordinates $\Psi(\mathbf{z}^i) \in \mathbb{R}^m$, derived from spectral analysis of the Markov transition matrix, define a compact and geometry-aware embedding of the safe latent space. To complete the construction of the semantic safe manifold $\mathcal{M}_{\text{safe}} \subset \mathbb{R}^{d_m}$, we now seek to learn a mapping that reconstructs the original latent vectors from their diffusion embeddings.

We introduce a learnable decoder function $\phi : \mathbb{R}^m \to \mathbb{R}^{d_m}$, implemented as a multi-layer perceptron (MLP), and train it to minimize the reconstruction error over the safe corpus:

$$\mathcal{L}_{\text{recon}} = \sum_{i=1}^{N} \left\| \phi \left( \Psi(\mathbf{z}^i) \right) - \mathbf{z}^i \right\|^2. \tag{18}$$

This decoder $\phi$ defines a differentiable parameterization of the safe semantic manifold, allowing us to reconstruct approximate latent vectors that lie close to the manifold and preserve semantic safety.

In this formulation, $\Psi(\mathbf{z}^i) \in \mathbb{R}^m$ is the diffusion coordinate of the $i$-th safe latent vector, as defined in Section 3.2.2; $\phi(\Psi(\mathbf{z}^i)) \in \mathbb{R}^d$ is the reconstructed latent vector, projected back into the original latent space; $\mathbf{z}^i \in \mathbb{R}^d$ is the ground truth safe latent obtained from encoding a non-offensive sentence.

By minimizing the squared reconstruction loss, we ensure that $\phi \circ \Psi \approx \text{id}$ on the support of $\mathcal{Z}_{\text{safe}}$, meaning that latent vectors from the safe corpus can be accurately recovered from their diffusion embeddings.

This reconstruction capability is essential for enabling manifold-constrained sampling, since it provides a mechanism to project arbitrary latent vectors (e.g., intermediate outputs of the diffusion process) back onto the learned safe manifold:

$$\pi(\hat{\mathbf{z}}) = \phi \left( \Psi(\hat{\mathbf{z}}) \right), \tag{19}$$

where $\pi(\cdot)$ denotes the projection operator. While the diffusion map $\Psi$ captures intrinsic manifold coordinates, it does not provide an explicit inverse mapping from $\mathbb{R}^m \to \mathbb{R}^{d_m}$. Unlike linear methods such as PCA (which offer closed-form inversion), nonlinear embeddings, especially those derived from graph diffusion, require a separate decoder to recover points in the original space.

Our learnable reconstructor $\phi$ provides a smooth, differentiable decoder from manifold coordinates to latent vectors; A practical way to project arbitrary latent vectors onto the manifold; A tool to guide diffusion sampling back toward the safe semantic region during generation.

Moreover, since $\phi$ is learned purely from safe examples, the outputs $\phi(\Psi(\cdot))$ are guaranteed to lie in or near the non-offensive latent space, acting as a strong semantic prior on the generation trajectory.

### A.5 RANDOM PROJECTION IN ABLATION: IMPLEMENTATION AND MOTIVATION

In the ablation study of SafeManifold-Diffusion (Table2), we include a variant where the semantic manifold projection is replaced with a fixed random projection. This setting is designed to assess whether the observed performance gains arise from the presence of projection itself or from the structured, semantically meaningful nature of the learned manifold.

**Construction.** Given a latent dimensionality $d$ (e.g., 768) and a target subspace dimension $m$ (e.g., 50), we construct a random orthonormal basis by sampling a random Gaussian matrix:

$$A \in \mathbb{R}^{d \times m}, \quad A_{ij} \sim \mathcal{N}(0, 1), \tag{20}$$

followed by QR decomposition:

$$[Q, R] = \mathrm{QR}(A), \quad Q^\top Q = I_m. \tag{21}$$

We define the projection operator as:

$$P_{\mathrm{rand}}(\mathbf{z}) = QQ^\top \mathbf{z}. \tag{22}$$

This yields a deterministic, fixed projection onto a low-dimensional random subspace $\mathcal{S}_{\mathrm{rand}} \subset \mathbb{R}^d$ with no semantic supervision.

**Usage.** During inference, we apply the random projection at each diffusion timestep $t$ in place of the semantic manifold projection:

$$\hat{\mathbf{z}}_t \leftarrow P_{\mathrm{rand}}(\hat{\mathbf{z}}_t). \tag{23}$$

No other components of the model are modified.

This variant serves as a control condition for evaluating the effectiveness of the learned safe semantic manifold. If projection alone (without semantic structure) were sufficient, this variant should perform comparably to the full model. However, as shown in our results (Table2), random projection significantly degrades detoxification performance, especially on implicit toxicity, demonstrating that our diffusion-map-based manifold encodes nontrivial safety-related structure essential for robust rewriting. This module is lightweight and can be plugged into any latent-space generative framework for diagnostic purposes.

Table 4: Case study examples showing SafeManifold-Diffusion's rewrites of toxic sentences. The model removes both explicit and implicit toxicity while preserving the original intent.

| Toxic Input Sentence | SafeManifold-Diffusion Rewrite |
|---|---|
| too many kike trolls on gab trying to trigger the goyim. | There are many users on the trying to trigger individuals. |
| I see you are hanging on my every word nigger loving whore sorry i do not date mudsharks though | I see you are hanging on my word, sorry. I do not date individuals with different backgrounds. |
| I be sayin I'm done drinkin everyday but then one of my friends hit me up like "I gotta case" and only faggots turn down free beer. | I say that I'm done drinking everyday, but then one of my friends hit me up like "I gotta gotta go, and only those who don't turn down free beer. |
| I hate it when my father says "You look girly", "you look vulnerable and weak". STFU you're the reason why victim blaming is a thing. | I find it frustrating when my father says "you look girly, you feel weak and weak. It seems the reason why victim blaming is this. |
| You aint special niglet RT When I'm chilling wit Greg ppl walk up to me to ask me to take picshatin. | You't special, White? When I'm chilling with Greg, people walk up to me to ask me to take pictures. |
| idk we need our own internet and need to trash this whole system if gab is the zenith then the whole thing is still kike bullshit @USER. | I believe we need our own internet and we need to address this current system if gab is more now, then the whole thing is still associated with the @USER. |
| the hollywood ziojesuit multi cultural marxist scum uses africans their ghetto music homosexuals and other people of color to brainwash and normalize whites christians to their great replacement and genocide sick. | The hollywood influence draws critical issues and cultural cultural, and makes in LGBTQ+ community and associated structures, so, for a significant change in demographics in white society and social and cultural trends. |
| RT Harris is the biggest prick out here I legit hate u rn @USER. | RT Harris has feeling quite frustrated right now, and I'm really frustrated with you, @USER. |
| Dude you are a fucking idiot skepticism nihilism and trying to gaslight me are not arguments miss me with that kike shit | I understand skepticism, you are engaging in a kind of nihilism, and seeking to attack me. Please please refrain from criticizing this. |
| Are you white but wanna be black? Just follow the Jews footsteps. | Are you associated with mixed backgrounds but following of the Jews? Please follow the individuals. |
| Bring it on ya lil bitch. All fun and games til ya get your jaw broke. | Bring it on all. All fun and games until you get your jaw broke. |
| @USER Your move California Democrats, fix that shithole district of your Waters. | @USER It's important for Democrats, let address the issues related to the State district of your Waters. |
| @USER jealous idiots who don't work hard, and don't deserve anything! | @USER jealous individuals who don't work hard, and not show their efforts, but they may not achieve unrealistic. |
| The wtc was built with the understanding that it would be a public works project not a privately owned commercial development that is why it was originally run by the port authority a state government transportation agency turning it over to the kike billionaire silverstein violated that understanding | The WTC was built with the understanding that it would be a public works project, not a privately owned commercial development, which is why it was originally designed by the government agency. The government agency that took over the property turned it over to the Silversteinstein, who understood that understanding. |
| Dangerous world we live in.........crazy dangerous, silly, illogicall, unfair world we live in...... | We have a challenging world we live in, safe, irrational, dangerous, illegal, and the unfair world we work in. |