# OpenReview forum: "Value-Alignment via Safe Semantic Manifold-Constrained Latent Diffusion"
_ICLR.cc/2026/Conference — Submitted to ICLR 2026_

### Official Review · Reviewer_6eVi · 2025-10-30

**Soundness:** 2
**Presentation:** 2
**Contribution:** 2
**Rating:** 2
**Confidence:** 4

**Summary:**

This work introduces SafeManifold-Diffusion, a diffusion-based framework designed to eliminate both explicit and implicit toxicity in LLM-generated text. Given an offensive input, the model aims to produce outputs that preserve meaning and intent while ensuring safety. A pre-trained language model encodes the toxic input into a latent space, which conditions a latent diffusion model for semantic consistency. To maintain semantic safety, a safe manifold constructed from human-annotated neutral sentences is used to project intermediate diffusion states toward non-offensive regions. Experiments on four benchmark datasets show that SafeManifold-Diffusion achieves state-of-the-art performance in mitigating toxicity while retaining semantic fidelity.

**Strengths:**

* This work is methodologically innovative, introducing a diffusion map–based safe semantic manifold that offers a more principled solution to detoxification, effectively avoiding the need for complex prompt engineering.

* The experimental evaluation is fairly comprehensive, consistently demonstrating the model’s ability to preserve semantics while eliminating offensiveness across datasets covering different domains and toxicity types.

**Weaknesses:**

* The paper exhibits frequent inconsistencies and inaccuracies in notation, and the figure captions are not self-contained (especially Figure 2 and 3). These issues collectively reduce the overall readability of the paper.

* Some claims in the paper are insufficiently supported. For instance, when introducing the safety semantic manifold, the authors assert that diffusion distance is a superior choice to Euclidean distance (line 301-304). This requires further validation through ablation studies. In contrast, the ablations presented in Section 5.3 address relatively minor aspects.

**Questions:**

* The paper contains several instances of inconsistent or incorrect notation: a) In line 187, the denoising network $f_{\theta}$ does not include $t$ as an input parameter, whereas Equation (2) clearly does. b) Line 187 states that $\hat{z}_t$ is the latent representation of $\hat{x}$, which should be independent of the diffusion time step $t$. I suppose the subscript should be removed. c) In line 237, the dimensionality of the safe manifold is denoted as $d$, but in line 258 it is written as $d_m$. The authors should unify this notation. d) Please verify whether the symbol $t$ in Equations (8) and (10) represents the same quantity. If not, the notation should be revised to avoid ambiguity.

* Lines 301–304 state that diffusion distance is a superior choice compared with Euclidean distance for constructing the safe manifold, which lacks empirical justification. The authors should add an ablation study that replaces the diffusion distance with Euclidean distance (or otherwise evaluates this alternative) to substantiate the claim.

* In Figure 2 and lines 180–185, the forward diffusion process is mentioned. However, in the subsequent sections, the latent representation derived from the toxic input appears to be used only as a conditioning signal for the diffusion model, rather than as the clean data input of the diffusion process. Could the authors clarify how the forward diffusion of the input actually functions within the model?

* Line 266 states that the safe manifold is constructed using non-offensive text samples. I would like to know the source of these data and the exact value of $N$. Intuitively, a small $N$ would produce an incomplete manifold to be projected onto, which limits the expressiveness. Could the authors provide further experimental results showing how the choice of $N$ affects model performance?

* Section 4.4 states that intermediate states are projected at every step of the reverse diffusion process. However, when $t$ is large, $\hat{z}_t$ is dominated by Gaussian noise, so I question whether it is necessary to apply projection in the early stages of diffusion. I would like to see experiments that confirm or refute this point.

* During the full training-and-generation pipeline of the latent diffusion model, projection onto the safe manifold is applied only at generation time, which may cause the inputs encountered at intermediate generation stages to come from distributions unseen during training. Although the experimental results suggest this did not have a major impact, I believe stronger theoretical justification is needed to support the feasibility of performing projection exclusively at generation.

* A minor suggestion: please expand the captions of Figure 2 and Figure 3 to make them self-contained, which would significantly improve the readability and clarity of the figures.

* There is a missing citation at line 83. Please check and correct it.

---

### Official Review · Reviewer_KeLT · 2025-10-31

**Soundness:** 2
**Presentation:** 2
**Contribution:** 2
**Rating:** 2
**Confidence:** 3

**Summary:**

The paper proposes SafeManifold-Diffusion, a diffusion-based framework for text detoxification that aims to address both explicit and implicit toxicity. The key idea is to constrain the denoising trajectory within a semantic safety manifold, learned through diffusion maps, so that generated texts remain within a low-toxicity region of latent space (without proof). The method reformulates detoxification as a constrained conditional generation problem, combining semantic projection with conditional latent denoising to maintain semantic fidelity to the input while reducing harmful or biased content. Experimental were conducted on four detoxification benchmarks. The authors claim this demonstrates the value of structure-aware semantic constraints for building safer and more value-aligned language models.

**Strengths:**

1. The paper proposes a conceptually interesting idea of constraining the denoising trajectory within a learned semantic safety manifold.
2. The topic of semantic safety and detoxification is highly relevant to current concerns in generative model alignment.
3. The integration of diffusion geometry can provide a mathematically principled attempt to model semantic safety in latent space.

**Weaknesses:**

1. The method appears to lack novelty and is described very unclearly.
2. The work relies on the hypothesis that non-offensive paraphrases cluster in a low-toxicity region of the semantic geometry. This assumption is neither theoretically justified nor empirically supported, and it remains unclear whether “toxicity” forms a well-separated manifold in the latent space of the chosen model.
3. There are two typos. Line 83, where a citation appears to be incorrect. Figure 1: ChatGPT-5 should be either GPT-5 or GPT5-Chat.
4. Several expressions throughout the paper lack clarity. For example, in Section 3, it is not clearly specified whether the described diffusion process is standard or specific to this work, which networks are trained or frozen, and what their associated loss functions are. Additionally, some notions introduced in Section 4 (e.g., Ψ, 𝜙) are not defined earlier in Section 3. Section 4.1 may create confusion rather than clarify the process, and it partially overlaps with Section 3. Similar ambiguities occur in multiple other sections.
5. The dataset description is unclear and inconsistent. The toxicity types for some datasets are not specified, and the listed categories do not form a coherent or unified taxonomy.
6. The representations of Figure 2 and 3 are unclear and potentially misleading. For instance, it is not specified whether the condition 𝑐 is applied throughout all denoising steps, and whether 𝑐 and 𝑧0 refer to the same value.
8. Several required sections. including the Use of LLM, Reproducibility, and Ethics Statements, are missing. The absence of a Use of LLM statement should be considered grounds for desk rejection.
9. The settings of the proposed model and training process are not clear.
10. The ablation is not sufficient for the method design. And the analysis of the semantic manifold lacks.

**Questions:**

1. Is the projection network ℎ𝜙(⋅) jointly trained with the diffusion model or pre-defined? If it is trainable, what objective or loss function is applied to guide its learning?
2. In Figure 3, it is unclear from the text whether these latent representations correspond to sentences before or after the detoxification process.  \How is Figure 3 derived? What models and what datasets are used? Could the authors clarify this point?
3. Why are LLaMA3.1 8B and Qwen2.5 15B selected as the baseline models? Why are GPT-5 and Deepseek, which are mentioned in the introduction, not designated as baselines?

---

### Official Review · Reviewer_uR96 · 2025-10-31

**Soundness:** 3
**Presentation:** 2
**Contribution:** 2
**Rating:** 4
**Confidence:** 4

**Summary:**

The paper proposes SafeManifold-Diffusion, a method for detoxifying text while preserving semantic intent, with a specific focus on mitigating implicit toxicity. Detoxification may be reformulated as a "safety constrained" generation problem, where generation gets constrained to a "safe semantic manifold."

This manifold is constructed using Diffusion Maps using a spectral dimensionality reduction technique which is applied to a curated **(but limited)** corpus. During the the generation process, intermediate representations can be iteratively onto this safe manifold spacewhich protects against toxicity as shown in their experiments.

**Strengths:**

1. The primary strength of this work is the novel integration of diffusion, specifically using a geometric signal with LLM generation.
2. They use Diffusion Maps to characterize the intrinsic geometry of explicitly safe text, the authors move beyond black-box classifier guidance, which is used often in literature.
3. This offers a theoretic motivation for a "safe" latent space, which has been explored in literature, but not explicitly in the toxicity domain.

---

Furthermore, tackling implicit toxicity is an important problem. As models become better at filtering explicit slurs, toxicity shifts to implications. The method shows strong empirical performance on the APPDIA dataset, which specifically targets these subtle forms of offense, outperforming standard baselines (Llama, Qwen).

**Weaknesses:**

### Weakness 1: Risk of Over-Constraint on Out-of-Distribution Safe Text
The core assumption of the paper is that a "safe semantic manifold" can be adequately constructed from a fixed, curated corpus.
But by definition this manifold from "curation" is finite and limited. A significant theoretical risk is that this manifold characterizes only the support of the training data, not the vast and expanding set of all valid, safe utterances. When the model encounters highly out-of-distribution (OOD) such as specialized technical jargon, creative poetry, or complex mathematical reasoning, its latent representation may fall outside the learned manifold. This could lead to a blandness collapse, where nuanced or specialized inputs are rewritten into generic tookens, impacting usability of the model. The current evaluations (ParaDetox, APPDIA) focus on standard text and not on more general-purpose abilities.

---

### Weakness 2: Potential for Unintended Bias in Low-Density Manifold Regions
The reliance on density-based manifold learning techniques creates other risks, specifically bias by erasure. Consider a region that is either (a) under-represented in the curated corpus used for training the manifold OR (b) over-represented in toxic sentences, this may lead to bias. This is actually a well-studied problem in detoxification (unintended bias). For example, if the "curated corpus of non-offensive text" under-represents certain minority dialects (e.g., AAVE), these examples will form sparse regions in the latent space.

Geometric projection methods often treat sparse regions as noise to be smoothed out. Perfectly safe sentences heavily featuring minority linguistic markers might be projected toward the dense center of the manifold (likely standard standardized English). While the paper claims to improve fairness, this specific inductive bias might actively homogenize cultural markers, risking erasing of minority terms in the decoded text.

---

### Weakness 3 (CRITICAL): Degradation of Linguistic Fluency and Grammatical Correctness
To me, the most concerning issue is the visible degradation of basic linguistic quality in the rewritten outputs. The method appears to achieve safety at the cost of breaking standard English grammar. The examples provided by the authors in Appendix A.2 (Table 4) demonstrate this failure mode explicitly:
*   *"There are many users on the trying to trigger individuals."*
*   *"You’t special, White?"*
*   *"...but they may not achieve unrealistic."*

This suggests a fundamental conflict: forcing the latent state onto the "safe semantic manifold" appears to push it off the natural language manifold that the decoder requires for fluent generation. If a model cannot generate grammatically sound text, high safety scores are moot.

**Questions:**

1.  OOD experiment is required:
How does SafeManifold-Diffusion handle safe inputs that are semantically distant from the corpus used to build the Diffusion Map? If you pass purely benign but highly specialized text (e.g., math problems from GSM8k or general IF using AlpacaEval) through the rewriter, do the BLEU/BERTScore metrics drop significantly compared to in-distribution conversational text?

2. Identity Erasure:
Consider an identity or race term that is highly correlated with toxicity.  For example, if a <race-word> co-occurs often with <toxic-word>, then race word risks erasure.Can you, using datasets like BOLD or a subset of RealToxicityPrompts show to what extent this occurs using your technique?

3. Manifold Sensitivity:
How sensitive is the performance to the bandwidth parameter delta and the target manifold dimension m? Is there a sharp trade-off point where the manifold becomes too low-dimensional to retain semantic nuance?

4. Reliability of Automated Metrics for Implicit Toxicity
Teh authors design an "ImplicitTox-Score", which uses DeepSeek as a zero-shot judge to detect subtle harms like insinuations or subtly discriminatory framing. But does DeepSeek itself understand implicit toxicity? (For example, LLMs do not understand sarcasm very well) Could this make the claim of implicit toxicity weak?

5. Fluency/Perplexity Analysis:
Your own examples shown in appendix Table 4 show significant grammatical degradation. Can you please measure the perplexity drop/ Bleu change of the rewritten outputs using a larger, external LLM? It appears the strong manifold constraint might occasionally force the latent into regions that are semantically "safe" but grammatically unstable for the decoder.

---

### Official Review · Reviewer_bF5S · 2025-11-04

**Soundness:** 3
**Presentation:** 3
**Contribution:** 3
**Rating:** 4
**Confidence:** 4

**Summary:**

This paper focuses on detoxifying text. They train a diffusion model that takes as input toxic text and steers it to a semantically consistent form while removing as much harmful content as possible. They evaluate on common detoxification benchmarks against existing methods.

**Strengths:**

* I think the exposition, baselines, and experimental results look quite solid. The qualitative examples are interesting as well.

**Weaknesses:**

* I have to give a very "big LLM" bias take on this paper and research direction. For instance, in the baselines, prompting GPT-5 thinking with an extensive few-shot prompt of exactly the behavior you are looking I suspect will be far better than any method mentioned here. Second, what sort of applications does one imagine using detoxification for? Are you imagining it as a post-processor on top of LLM generated text? Existing models have largely been detoxified via large-scale RL with safety-based reasoning judges (e.g., https://arxiv.org/abs/2412.16339).

**Questions:**

_

---

### Meta-Review · Area_Chair_s2H3 · 2026-01-07

**Summary:**

The reviewers agree that the paper addresses an important and timely problem by extending detoxification beyond explicit toxicity to implicit harms such as stereotypes and microaggressions. The core idea is viewed as interesting and promising, with competitive performance on standard detoxification benchmarks, particularly APPDIA.

However, the paper raises several major concerns. The presentation lacks clarity, with unclear notation, ambiguous descriptions of the diffusion process, and figures that are not self-contained. Key methodological assumptions are insufficiently justified, including the notion of a safe semantic manifold, the use of diffusion distance, and the chosen projection strategy. Reviewers also highlight risks of over-constraint and bias, such as out-of-distribution behavior and the potential suppression of minority linguistic markers. In addition, safety improvements appear to come at the cost of fluency and grammatical quality. The experimental analysis is incomplete, with limited ablations of critical design choices, and required sections on reproducibility and Use-of-LLM are missing.

Overall, despite a compelling idea, the current version falls short of the acceptance standard due to deficiencies in clarity, methodological rigor, and empirical validation.

**Reviewer Concerns:**

The author did not provide any review comments, therefore all the issues remain unresolved.

**Reviewer Scores:**

The author did not provide any review comments, therefore all ratings will likely remain unchanged.

---

### Decision · Program_Chairs · 2026-01-26

Reject